# Health professionals' technology readiness on the acceptance of teleradiology in the Amhara regional state public hospitals, northwest Ethiopia: Using technology readiness acceptance model (TRAM)

Araya Mesfin Nigatu[1]*, Tesfahun Melese Yilma[1], Lemma Derseh Gezie[2], Yonathan Gebrewold[3], Monika Knudsen Gullslett[4], Shegaw Anagaw Mengiste[5], Binyam Tilahun[1]

1 Department of Health Informatics, Institute of Public Health, University of Gondar, Gondar, Ethiopia, 2 Department of Epidemiology and Biostatistics, Institute of Public Health, University of Gondar, Gondar, Ethiopia, 3 Department of Radiology, School of Medicine, University of Gondar, Gondar, Ethiopia, 4 Faculty of Health and Social Sciences, University of South-Eastern Norway, Drammen, Norway, 5 Management Information Systems, University of South-Eastern Norway, Drammen, Norway

* hitdt2005@gmail.com

## Abstract

### Background

Considering individual differences caused by personality differences is crucial for end users' technology acceptance. However, previous studies overlooked the influence of users' technology readiness on technology acceptance. This study, therefore, aimed to evaluate the influence of technology readiness on teleradiology acceptance in the Amhara Regional State Public Hospitals using a technology readiness acceptance model.

### Methods

An institutional-based cross-sectional mixed study design was conducted in September 2021 among 547 health professionals working at sixteen public hospitals in the Amhara region of northwest Ethiopia. Eight key informants were interviewed to explore organizational-related factors. Face-to-face and Google Meet approaches were used to collect the data. We applied structural equation modeling to investigate the influence of technology readiness on health professionals' teleradiology acceptance using Analysis of Moment Structures Version 23 software.

### Results

Of the total participants, 70.2% and 85.7% were ready and intended to use teleradiology, respectively. According to technology readiness measuring constructs, optimism and innovativeness positively influenced health professionals' technology acceptance. Perceived ease of use and perceived usefulness showed a statistically positive significant effect on

**Data Availability Statement:** All relevant data are uploaded as supporting information (S1 dataset and S2 audio records).

**Funding:** The work was supported financially by the University of Gondar under award number (Ref. No: 05/2886/2013). The funder, however, played no role in the study design, data collections, analysis of results, interpretation of data, writing of the manuscript, nor the decision to submit the manuscript for publication.

**Competing interests:** The authors declared no potential conflicts of interest with respect to the research, authorship, and/or publication of this article.

**Abbreviations:** AMOS, Analysis of Moment Structure; CI, confidence Interval; eHealth: Electronic Health; FMOH, Federal Ministry of Health; ICT, Information and Communication Technology; IT, Information Technology; RHB, Regional Health Bureau; SEM, Structural Equation Modeling; TAM, Technology Acceptance Mode; TR, Technology Readiness; TRAM, Technology Readiness and Acceptance Model.

health professionals' intention to use teleradiology. In addition, a statistically significant mediation effect was observed between technology readiness measuring constructs and behavioral intention to use. Furthermore, a shortage of budget, inadequate infrastructure, and users' lack of adequate skills were reported as critical organizational challenges.

## Conclusions

We found a higher proportion of readiness and intention to use teleradiology among health professionals. Personality difference measuring constructs and organizational factors played considerable influence on teleradiology acceptance. Therefore, before the actual implementation of teleradiology, ensuring the system's user-friendliness, improving infrastructure, allocating an adequate budget, and availing of capacity-building opportunities are recommended.

## Introduction

Imaging holds immense importance within healthcare systems due to its critical role in patient treatment through image-based interventions and diagnoses [1]. Currently, numerous online platforms offer imaging services [2]. Of this, teleradiology, an emerging digital health technology, has garnered significant attention from countries due to its substantial benefits to patients and physicians [3–5]. The technology facilitates remote image interpretation and consultation between medical experts, overcoming geographical barriers [6]. Implementing teleradiology has numerous advantages, such as enabling healthcare providers to provide timely diagnosis, to avoid patient travel, to reduce out-of-pocket travel expenses, to decrease radiology report waiting time, to avail the service at all the times (24/7/365) [7], and to reduce patient transfer [7, 8]. However, its adoption is limited, particularly for low-income countries (LICs) [1, 5] because of socio-political systems [5], low penetration of information technology (IT) in the healthcare industry compared to non-health sectors [9], lack of adequate infrastructure, high initial costs [10], lack of technical experts, and system simplicity [11].

The Ethiopian Federal Ministry of Health has formulated a digital blueprint action plan [12], which includes teleradiology as a short-term pilot program to improve access and equity in remote healthcare services. While there are studies available on health professionals' readiness for various eHealth technologies [13–16], research focusing on the influence of personality differences on technology acceptance is lacking. Nevertheless, it is essential to accommodate individual differences when evaluating users' intentions to adopt new technology [17, 18]. Several theoretical models have been proposed to explain individuals' intentions to use technology [19, 20]. Among these models, the Technology Acceptance Model (TAM), derived from the Theory of Reasoned Action (TRA) [21], has gained significant popularity as a framework for studying the acceptance of new technology [22]. The Technology Acceptance Model (TAM) has limitations in addressing individual differences [23] and fails to explain the impact of external variables on users' perceived usefulness and perceived ease of use [24]. To address this gap, the current study utilized the Technology Readiness and Acceptance Model (TRAM), which incorporates dimensions of personality and technology acceptance to account for individual differences in explaining users' intention to adopt new technology in their work environment [25, 26]. Additionally, the study employed a qualitative approach to complement

the quantitative aspect by addressing organizational barriers. The study aimed to answer the following two questions:

a) How health professionals' technology readiness influences the acceptance of teleradiology?

b) What are the organizational related barriers influencing the adoption of teleradiology?

The study's description has relevance for both educational and administrative applications. The correlation between technology readiness and technology acceptance constructs can assist healthcare professionals in identifying mental factors that influence the inclination to adopt new healthcare technologies, potentially impacting the quality of radiology services. Furthermore, understanding the organizational barriers associated with the intention to use teleradiology can aid administrative bodies at various levels in their planning and evaluation processes.

The structure of this paper is as follows: The next section provides a literature review on Technology Readiness (TR) and the Technology Acceptance Model (TAM), which serves as the foundation for developing a model that examines the relationship between TR and each construct of TAM in the context of teleradiology. The third section focuses on the development of hypotheses. In the fourth section, the research methods and materials are described. The fifth section presents the results of the study. Finally, the sixth section includes the discussion and conclusion.

## Theoretical models and hypothesis development

**Theoretical models.**  The research framework was formulated using the TRAM as its foundation. Within the TRAM, the TR (Technology Readiness) factor plays a crucial role in shaping individuals' perceptions regarding the ease of use and usefulness of new technologies. These perceptions, in turn, have a significant impact on the acceptance and adoption of those technologies [27]. Parasuraman [28] introduced the technological readiness (TR) model. The TR model refers to "people's propensity to embrace and use new technologies for accomplishing goals for home life and at work". According to Parasuraman, the model has four personality difference measuring dimensions, namely optimism, inventiveness, discomfort, and insecurity. Parasuraman defined the four TR personality differences as: 1) optimism is "a positive view of technology and a belief that technology offers people increased control, flexibility, and efficiency in their lives"; 2) innovativeness is "a tendency to be an early adopter of technology and an opinion leader"; 3) discomfort is "a perception of being unable to control the technology and a feeling of being overwhelmed by it"; and 4) insecurity is "suspect of technology and doubt about its capability to work" [28].

From the four personality difference measuring domains, the first two (optimism and innovativeness) are considered enablers (positive drivers encouraging end-users to use the technology and to hold a positive attitude towards the technology), while the remaining two (discomfort and insecurity) are inhibitors (negative drivers making end-users reluctant to use technology) [29, 30]. A strong presence of enablers contributes to a higher overall readiness level, while the presence of inhibitors lowers the overall readiness level [28]. Individuals who are optimistic and innovative but experience less discomfort and insecurity are more likely to embrace and use new technologies [9]. The TR model is currently the most comprehensive measure of technology readiness for investigating health professionals' behavioral intentions to adopt new technologies [31].

On the other hand, the technology acceptance model (TAM) has been one of the most influential models for technology acceptance [32], is adapted from the theory of reasoned action (TRA) [33]. TAM consists of two key components (perceived ease of use and perceived utility), both of which have a significant influence on explaining the differences in user

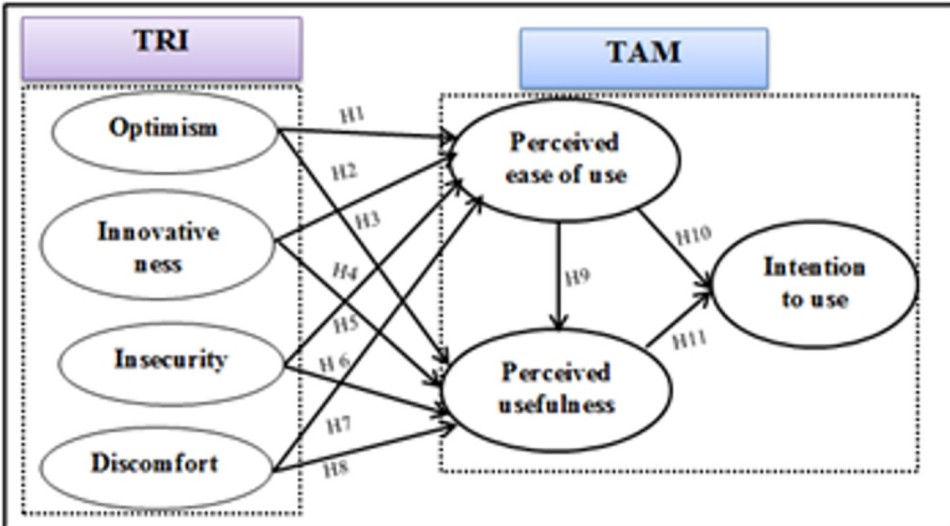

**Fig 1. Technology readiness and acceptance model adopted from previous literatures [29, 39, 40].**

intentions to use new technologies [32, 34]. Davis [32] defined perceived usefulness as "the degree to which a person believes that using a particular system would enhance his or her job performance." Whereas perceived ease of use as "the degree to which a person believes that using a particular system would be free of effort." The model has become well known as robust, powerful, and parsimonious for predicting user acceptance [35]. The model is also widely accepted in healthcare settings because it is an important framework for evaluating health professionals' overall intention to use new technology [34, 36].

**Hypothesis development.** According to previous research studies [18, 37–40], individual characteristics influence the relationships between perceptions and intentions to use technology. Hence, the context of the teleradiology system, the following hypotheses were proposed (Fig 1).

*Effect of "optimism" on the perceived usefulness and perceived ease of use*. Optimism is a construct that reflects people's positive attitude and intention to embrace and utilize new technology [18]. Optimistic individuals perceive technology as highly valuable and user-friendly, and they are less likely to dwell on the negative outcomes associated with it [18, 29]. Based on previous research [41, 42], it can be observed that optimists generally exhibit positive attitudes towards new technologies and display a higher level of enthusiasm for adopting them. Building on this observation, we formulated the hypothesis:

H1: Optimism positively affects the perceived ease of use of teleradiology.

H2: Optimism positively affects the perceived usefulness of teleradiology.

*The effect of "innovativeness" on the perceived usefulness and perceived ease of use*. Innovativeness significantly influences the perceived suitability and usefulness of new technologies [28]. Individuals with high innovativeness demonstrate a greater intrinsic motivation to embrace and explore new technologies [18]. They tend to be early adopters and perceive the new technology has lower complex [9]. Based on these observations, we propose the following hypothesis:

H3: Innovativeness positively affects the perceived ease of use of teleradiology.

H4. Innovativeness positively affects the perceived usefulness of teleradiology.

*The effects of "insecurity" on perceived ease of use and perceived usefulness.* Insecure individuals often lack self-confidence in the safety of new technologies and may perceive certain risks associated with their use [43]. Previous research suggests that insecurity is associated with lower levels of perceived usefulness and perceived ease of use [44]. Thus, we hypothesize:

H5. Insecurity negatively affects the perceived ease of use of teleradiology.

H6. Insecurity negatively affects the perceived usefulness of teleradiology.

*The effect of "discomfort" on the perceived ease of use and perceived usefulness.* Discomfort is the perception of a lack of control and feeling overwhelmed when using technology [18, 44]. As a result, individuals experiencing discomfort tend to view technology as complex and difficult to use [42]. Thus, we hypothesize:

H7. Discomfort negatively affects the perceived ease of use of teleradiology.

H8. Discomfort negatively affects the perceived usefulness of teleradiology.

*The effect of perceived ease of use on perceived usefulness.* Perceived usefulness is influenced by the perceived ease of use, with easier systems being considered more useful [8]. This suggests that user-friendly applications may be perceived as useful, but not all useful applications are necessarily user-friendly [41]. Based on these observations, we propose the following hypothesis:

H9. Perceived ease of use positively affects the perceived usefulness of teleradiology.

*The effects of perceived ease of use and perceived usefulness on behavior intention.* Perceived usefulness and ease of use influence technology adoption [29, 32]. Based on the Technology Acceptance Model (TAM), both factors impact an individual's intention to use technology. Therefore, we hypothesize:

H10: Perceived ease of use positively affects the behavior intention in using teleradiology.

H11. Perceived usefulness positively affects the behavior intention in using teleradiology.

## Methods and materials

### Study design and period

An institutional-based cross-sectional mixed study design was undertaken among healthcare professionals at public hospitals in the Amhara region of northwest Ethiopia to determine the influence of technology readiness on teleradiology acceptance. The study used a qualitative component to supplement the quantitative part since the TRAM model can't accommodate organizational factors [45]. The study was conducted in September 2021.

### Study setting

The Amhara region is located in the Northwestern and North Central parts of Ethiopia. According to the Amhara regional state 2021/2022 annual report, the region has a total of 19 Zones, including the city administrations, 181 districts, and 238 kebeles (the lowest administrative units). According to the report, in the region, there are 98 public hospitals, 917 health centers, and 3,725 public health posts. Moreover, there are 241 specialists, 1307 general practitioners, and 260 radiology professionals (83 radiographers, 139 first-degree radio-technologists, and 38 radiologists) available to serve an estimated 22.5 million people in the region.

This study was carried out at sixteen public hospitals found in two zones and one city administration of Amhara region.

## Participants and sample size determination

We included all health professionals, namely: radiologists, radiographers, internists, general practitioners, pediatricians, orthopedicians, gynecologists, emergency surgeons, ophthalmologists, dentists, anesthetists, and dermatologists, for the quantitative part of the study. As a result, the chance of selection bias was eliminated. We selected these participants based on their theoretical exposure to telemedicine and teleradiology in their graduate and undergraduate classes. Similarly, we selected key informants from Zones, region, and ministry of health for the qualitative part. The sample size for the quantitative part was determined by using 5:1ratio (5 observations per estimated parameter) [46]. Therefore, by considering the 97 free parameters of the model and the 10% non-response rate, we obtained the required sample size of 534. Though we calculated the minimal sample size, we used the maximum sample size of 569 by including all healthcare workers listed above from the selected hospitals. For the qualitative part, eight study participants were selected using maximum variation technique. The level of saturation was considered to determine the minimum sample size.

## Sampling procedure

Two zones and one city administration were selected using a simple random sampling technique. The study included all hospitals in the selected zones and city administrations. Moreover, all eligible health professionals from each public hospital were included in this study. Participants for the qualitative part were selected using a purposive sampling technique.

## Data collection tool development

The study employed TRAM adapted from previous studies [29, 37]. The questionnaire was divided into four parts. The first part of the questionnaire was socio-demographic characteristics; the second was the eHealth literacy scale; the third was experience-related; and the fourth was readiness and acceptance. The TR part included four constructs (optimism, innovativeness, discomfort, and insecurity) adopted from Parasuraman's instrument [12], which helped to measure health professionals' readiness for teleradiology implementation. On the other hand, the TAM part has three constructs (perceived ease of use, perceived usefulness, and behavioral intention) adapted from the Davis instrument [18], which enabled it to measure health professionals' teleradiology acceptance. We pre-tested the validated instrument outside the research area with similar populations' characteristics.

   We carried out content and face validity with 20 domain experts invited from the fields of public health, medicine, health informatics, and computer science. Experts were purposely selected considering their research experience and study content proximity. The item content validity index (I-CVI), scale content validity index (S-CVI), and Kappa coefficient were computed. As a result, one item was modified and two items were dropped based on the minimum acceptable value (Kappa coefficient $\geq$ 0.6, I-CVI $\geq$ 0.78, and S-CVI $\geq$ 0.8) recommended by previous studies [32, 45]. We performed forward and backward translations to ensure the consistency of the instrument. Finally, 35 items were included in our final TRAM model (seven items for optimism, seven items for innovativeness, eight items for discomfort, three items for insecurity, three items for perceived ease of use, four items for perceived usefulness, and three items for intention to use). The instrument, originally developed in English, was subsequently translated into Amharic, the local language. Eight health informatics professionals, who were

trained specifically for this purpose, conducted a self-administered data collection at each participating public hospital.

For the qualitative part, a literature review and behavioral science expert consultation were conducted to develop the semi-structured interview guide. The interview guide was written in English, then verified for face validity by experts, and then translated into the local language (Amharic) for administration.

## Study variables and operational definitions

In our model, we have three endogenous (dependent) and four exogenous (independent) variables. In SEM, an endogenous variable is any latent variable that is predicted by other variables (has at least one arrow leading into it from another latent variable) [47]. On the other hand, an exogenous latent variable is any latent variable that does not have an arrow pointing to it [47]. In this study, perceived usefulness, perceived ease of use, and intention to use were endogenous variables, whereas optimism, innovativeness, discomfort, and insecurity were exogenous variables.

Technology readiness is defined as "health professionals tendency to embrace and use technology' (in our case, teleradiology)" [39, 48]. We used TR constructs to evaluate the direct and indirect influence on the endogenous variables. On the other hand, the TAM construct of perceived usefulness is "the degree to which a person believes that using a particular system would enhance his or her job performance" [32]. Similarly, perceived ease of use is "the degree to which a person believes that using a particular system would be free of effort" [32]. Moreover, intention to use is 'the user's desire to use technology (in our context, teleradiology) in the future' [32, 49].

All constructs were measured using a five-point Likert scale ranging from score of '1-strongly disagree' to '5-strongly agree'. To compute the proportion of readiness, all the four item scores were summed up and divided by the number of items to create a composite variable scale (ranging from score 1 to 5). Finally, a two-point scale was generated by merging responses for 1, 2, and 3 into one category and the remaining 4 and 5 into another category [50–52]. Accordingly, the final score above three (agree and strongly agree) was categorized as 'Ready' while those final scores of three and below (neutral, disagree, and strongly disagree) as 'Not-Ready'. We used a similar approach to compute the proportion of intention to use teleradiology.

## Data collection procedures and quality control measures

The qualitative data were collected using a self-administered and structured pre-tested Amharic version questionnaire. The questionnaire was initially developed in English, then translated to Amharic for appropriateness in approaching study participants, and then translated back to English to ensure consistency. Two supervisors and eight data collectors participated in the data collection process. The research team was trained for two days to ensure that they understand the study's objective, the research tool, and the data collection procedure.

We used an interview guide to collect the qualitative data. The point of saturation was considered to reach the final sample size. The principal investigator approached key informants (medical directors, hospital managers, expert from the regional health bureau, expert from the federal ministry of health, and expert from Ethiopian Telecommunication Corporation) face-to-face and via the Zoom platform. All interviews were audio-recorded and lasted 42 to 60 minutes.

## Data management and analysis

We used EpiData version 3.1 for data entry and exported it to SPSS version 26 for descriptive analysis. Descriptive analysis such as mean and percentage was used to describe the demographic characteristics, readiness, and intention levels of health professionals towards teleradiology. We performed structural equation modeling (SEM) using AMOS version 23 to determine the degree of the relationship between exogenous and endogenous variables. Chi-square Ratio, Normal Fit Index (NFI), Goodness of Fit Index (GFI), Adjusted Goodness of Fit Index (AGFI), and Root Mean Square Error of Approximation (RMSEA) fit measure indices were used for checking the overall goodness of the model.

The path coefficients were interpreted as SEM coefficients, and the bootstrap method was used to assess the statistical significance of the path coefficients of predictors. $R^2$ represents the proportion of variance in the endogenous constructs that can be explained by the predictors. The statistical significance of predictor path coefficients with a p-value less than 0.05 was determined using a 95% bias-corrected confidence interval (CI) bootstrap technique. Finally, the p-value and standardized path coefficients were used to evaluate the relationship between endogenous and exogenous variables.

## Reliability of the data

The internal consistency of constructs was determined by Cronbach's alpha reliability coefficients. Literatures recommends that the minimum recommended test result of Cronbach's alpha value be greater than 0.6 [53, 54]. In this study, we computed Cronbach's alpha value for all of the seven constructs. Optimism ($\alpha$ = 0.91), innovativeness ($\alpha$ = 0.82), discomfort ($\alpha$ = 0.70), insecurity ($\alpha$ = 0.69), perceived ease of use ($\alpha$ = 0.66), perceived usefulness ($\alpha$ = 0.92), and intention to use ($\alpha$ = 0.89). However, the instrument's overall reliability on the 35 items has also shown a remarkable consistency of responses ($\alpha$ = 0.85). Therefore, the findings of our study satisfy the minimum acceptable criteria.

## Ethical approval and consent to participate

This study was approved by the Institutional Ethical Review Board (IRB) of the University of Gondar (Ref. No: VP/RTT/05/2554/2021). In addition, a support letter was secured from the Amhara Region Public Health Institute. The study participants were given an Amharic version of the information sheet that explained the purpose of the research and the procedure. They were also made aware that their participation was completely voluntary and that they do have the right to decline or withdraw at any time during the data collection process. All responses were kept confidential and anonymous. They were also informed that their participation was entirely voluntary and that they might decline or withdraw at any time. All responses were kept confidential and anonymous (i.e., codes were assigned instead of potentially identifiable records). Written informed consent was obtained from each study participant.

## Results

### Demographic characteristics

A total of 547 valid questionnaires were collected, yielding a response rate of 96.1%. The median age of the study participants was 29 years (IQR: 32–27 years), and 334 (61.1%) of the participants were within the age category of 26–30 years. Above three-fourths of the respondents, 488 (89.2%) were males. Nearly half of the respondents, 262 (47.9%) were married and 481 (87.9%), were Orthodox Christians by religion. Regarding academic qualification, 222

**Table 1. Demographic characteristics of health professionals in the Amhara regional state public hospitals, north-west Ethiopia, 2021 (n = 547).**

| Variables | Category | Frequency (%) |
|---|---|---|
| Age in years Median ± IQR [29 (IQR: 32–27)] | < = 25 | 53 (9.7) |
| | 26–30 | 334 (61.1) |
| | 31–35 | 116 (21.2) |
| | 36–40 | 32 (5.9) |
| | 41–45 | 6 (1.1) |
| | > = 46 | 6 (1.1) |
| Gender | Male | 488 (89.2) |
| | Female | 59 (10.8) |
| Religion | Orthodox | 481 (87.9) |
| | Muslim | 43 (7.9) |
| | Protestant | 21 (3.8) |
| | Others [a*] | 2 (0.4) |
| Marital status | Married | 262 (47.9) |
| | Single | 267 (48.8) |
| | Divorced | 7 (1.3) |
| | Widowed | 2 (0.4) |
| | Separated | 9 (1.6) |
| Educational status | Diploma | 37 (6.8) |
| | First Degree | 58 (10.6) |
| | Masters | 47 (8.6) |
| | Medical doctors | 222 (40.6) |
| | MD+(specialty/subspecialty) | 84 (15.4) |
| | Residents | 99 (18.1) |
| Department | Internal Medicine | 150 (27.4) |
| | Surgery | 104 (19.0) |
| | Pediatrics | 96 (17.6) |
| | Orthopedics | 17 (3.1) |
| | Gynecology obstetrics | 92 (16.8) |
| | Radiology | 63 (11.5) |
| | Others [c*] | 25 (4.6) |

Others

[a*] = Catholic; Others

[b*] = Adult OPD, Anesthesia, Emergency, dentistry

(40.6%) of participants were general practitioners, out of which 150 (27.4%) were working in the internal medicine department during the data collection period (Table 1).

## Teleradiology readiness

We assessed teleradiology readiness by the four technology readiness measuring constructs, namely optimism (seven items), innovativeness (seven items), discomfort (eight items), and insecurity (three items). This study's results indicated that 384 (70.2% [95% CI: 66.2%–74.0%]) of health professionals were ready to use teleradiology.

## Health professions' eHealth literacy

The majority of the health professionals 334 61.1%) knew what health resources were available on the Internet. Similarly, more than half of them, knew where to find resources 314 (57.4%)

**Table 2. eHealth Literacy status among health professionals in the Amhara regional state public hospitals, northwest Ethiopia, 2021 (n = 547).**

| Items | SD # (%) | Disagree # (%) | Neutral # (%) | Agree # (%) | SA # (%) |
|---|---|---|---|---|---|
| I know what health resources are available on the Internet. | 9 (1.6) | 12 (2.2) | 35 (6.4%) | 334 (61.1) | 157 (28.7) |
| I know where to find helpful health resources on the Internet. | 6 (1.1) | 24 (4.4) | 52 (9.5) | 314 (57.4) | 151 (27.6) |
| I know how to find helpful health resources on the Internet. | 7 (1.3) | 24 (4.4) | 42 (7.7) | 317 (58.0) | 157 (28.7) |
| I know how to use the Internet to get an answer for my questions about health. | 9 (1.6) | 24 (4.4) | 64 (11.7) | 277 (50.6) | 173 (31.6) |
| I know how to use the health information I find on the Internet to help me. | 7 (1.3) | 21(3.8) | 52 (9.5) | 290 (53.0) | 177 (32.4) |
| I have the skills to evaluate the health resources I find on the Internet. | 18 (3.3) | 36 (6.6) | 91(16.6) | 267 (48.8) | 135 (24.7) |
| I can tell high quality health resources from low quality health resources on the Internet. | 10 (1.8) | 62 (11.3) | 129 (23.6) | 230 (42.0) | 116 (21.2) |
| I feel confident in using information from the Internet to make health decisions. | 17 (3.1) | 44 (8.0) | 88 (16.1) | 278 (50.8) | 120 (21.9) |

SD: strongly disagree; SA: Strongly Agree

#: frequency

and how to find them 317 (58.0%). Moreover, 277 (50.6%) and 278 (50.8%) of the health professionals knew how to use the internet to get an answer to their questions about health and were confident in using information from the Internet to make health decisions, respectively. However, 267 (48.8%) and 230 (42.0%) of the health professionals had the skills to evaluate the health resources and to identify the quality resources below the average respectively (Table 2).

## Intention to use teleradiology

We assessed the intention to use teleradiology among health professionals by three items. The overall reliability test (Cronbach alpha) of the intention to use the teleradiology construct was 0.89. About 469 (85.7% [95% CI: 82.8%– 88.7%]) of the study participants had the intention to use teleradiology.

## Structural model assessment and hypotheses testing

We used a 95% bias-corrected CI bootstrap technique to determine the statistical significance of predictor path coefficients. $R^2$ and standardized coefficient values were used to interpret the variability and effects of the endogenous variables, respectively. The study constructs together explained 54.5% of the variance in health professionals' intention to use teleradiology (Fig 2).

## Path coefficients and hypothesis testing

The influence of exogenous variables on endogenous variables was declared using a cutoff point of 95% bias-corrected confidence interval (CI) and a p-value less than 0.05. The key influencing factors optimism, innovativeness, perceived ease of use, and perceived usefulness were interpreted by the path coefficients of the constructs. To test the direct and indirect effects of exogenous variables on endogenous variables, the structural model was evaluated by producing 95% bias-corrected confidence interval (CI) bootstrap values.

We found that optimism has a positive and significant direct influence on perceived ease of use [β = 0.58, 95% CI: 0.456–0.697] and perceived usefulness [β = 0.29, 95% CI: 0.137–0.445]. Similarly, a significant positive influence was observed between innovativeness and perceived ease of use [β = 0.13, 95% CI: 0.016–0.242]. Furthermore, perceived ease of use has a significant positive influence on perceived usefulness [β = 0.53, 95% CI: 0.368–0.685] and behavioral intention to use [β = 0.16, 95% CI: 0.003–0.330]. Likewise, perceived usefulness showed a statistically significant positive influence on behavioral intention to use [β = 0.61, 95% CI: 0.454–0.761]. However, neither discomfort nor insecurity readiness measuring constructs showed a

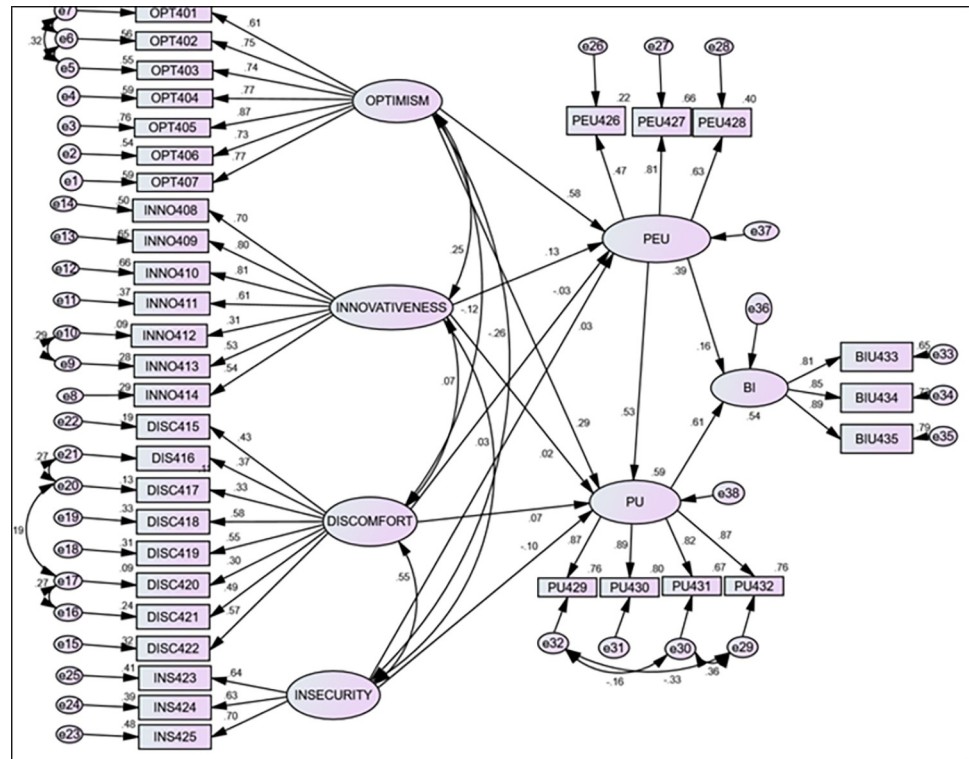

**Fig 2. Structural model with standardized path coefficients.**

statistically significant effect on perceived ease of use or perceived usefulness (Table 3 and Fig 2).

## Structural equation modeling model fits

The study findings indicated that the model was regarded as a well-fitting model as it satisfied the minimum recommended SEM model fit indices cut-offs (Table 4).

## Effect of mediators on intention to use teleradiology

We carried out a mediation analysis using AMOS with bootstrap at a 95% bias-corrected confidence interval to test the indirect influence of technology readiness measuring constructs on behavioral intention to use teleradiology through perceived usefulness and perceived ease of use.

Optimism had a greater indirect effect on perceived usefulness ($\beta$ = 0.325) when mediated by perceived ease of use than its direct effect ($\beta$ = 0.292). Similarly, the effect of perceived ease of use on behavioral intention was nearly three times higher when mediated by perceived usefulness ($\beta$ = 0.417) compared to its direct effect ($\beta$ = 0.162). Moreover, innovativeness showed a significant indirect influence on behavioral intention when mediated by perceived ease of use [$\beta$ = 0.081) (Tables 3 and 5).

## Qualitative data results

We applied a qualitative approach to explore organizational barriers affecting teleradiology adoption. Key informants reported that, most of the health organizations were eager to adopt

**Table 3. Structural equations modeling and hypothesis testing among health professionals in the Amhara regional state public hospitals, northwest Ethiopia, 2021 (n = 547).**

| Causal path | Hypothesis | Estimate (β) | P-value | 95% CI | | Decision |
|---|---|---|---|---|---|---|
| | | | | Lower | Upper | |
| OPT → PEU | H1 | 0.581 | 0.001* | 0.456 | 0.697 | Accepted |
| OPT → PU | H2 | 0.292 | 0.004* | 0.137 | 0.445 | Accepted |
| INNO → PEU | H3 | 0.130 | 0.025* | 0.016 | 0.242 | Accepted |
| INNO → PU | H4 | 0.016 | 0.693 | -0.067 | 0.096 | Rejected |
| INS →PEU | H5 | 0.034 | 0.641 | -0.133 | 0.215 | Rejected |
| INS → PU | H6 | -0.100 | 0.177 | -0.37 | 0.043 | Rejected |
| DISC → PEU | H7 | -0.029 | 0.721 | -0.215 | 0.141 | Rejected |
| DISC → PU | H8 | 0.067 | 0.299 | -0.057 | 0.193 | Rejected |
| PEU → PU | H9 | 0.529 | 0.001* | 0.368 | 0.685 | Accepted |
| PEU → BI | H10 | 0.162 | 0.044* | 0.003 | 0.330 | Accepted |
| PU→BI | H11 | 0.613 | 0.001* | 0.454 | 0.761 | Accepted |

Source: AMOS own analysis output

OPT: Perceived Ease of Use; INNO: Innovativeness; DIS: Discomfort; INS: Insecurity; PEU: perceived ease of use; PU: Perceived usefulness; BI: Behavioral intention; CI: Confidence Interval

teleradiology to successfully deliver the radiology service. Our study explored factors that could affect teleradiology implementation. Of these, shortage of budget, frequent power outages, low bandwidth of internet connection, users' limited computer skills, and lack of technical experts were the critical challenges reflected by the key informants. A key informant stated:

> "*Implementing technology is critical to safely delivering healthcare services in a reasonable amount of time. Users' attitudes towards technology use have shifted dramatically. Currently, let alone for their healthcare facilities, they prefer restaurants and café with Wi-Fi service availability to accomplish their day-to-day activities. However, a high shortage of budget is the major challenge to upgrading the existing internet bandwidth.*" (Hospital Medical Director, 2 years of working experience)

The study confirmed that the internet service provider's readiness to offer the service on time was promising. A key informant from the Ethiopian Telecommunication Corporation stated:

**Table 4. Model fitness for technology readiness on teleradiology acceptance among health professionals in the Amhara regional state public hospitals, northwest Ethiopia, 2021 (n = 547).**

| Fit indices | Threshold Value | Authors | Current study results | Conclusion |
|---|---|---|---|---|
| Parsimonious fit (Ratio to Chi Square) | ≤ 3 | Schermelleh-Engel et al., 2003 [55] | 2.30 | Accepted |
| Goodness-of-index (GFI) | > 0.9 | Shevlin et al., 1998 [56] | 0.89 | Accepted |
| Adjusted goodness-of-index (AGFI) | > 0.8 | Baumgartner et al., 1996 [57] | 0.87 | Accepted |
| Comparative fit index (CFI) | > 0.9 | Bentler et al., 1980 [58]; Hu et al., 1999 [59] | 0.92 | Accepted |
| Tucker–Lewis index (TLI) | > 0.9 | Bentler et al., 1980 [58] | 0.911 | Accepted |
| Root mean square error of approximation (RMSEA) | < 0.05 | Bentler et al., 1980 [58]; Barrett, 2007 [60] | 0.049 | Accepted |

Source: AMOS analysis output

**Table 5. Standardized indirect effects of technology readiness on teleradiology acceptance among health professionals in the Amhara Region public hospitals, northwest Ethiopia, 2021 (n = 547).**

| Path | Mediator | Estimate (β) | 95% CI | | P-value |
|---|---|---|---|---|---|
| | | | Lower | Upper | |
| Optimism → Perceived Usefulness | PEU | 0.325 | 0.205 | 0.497 | < 0.001 |
| Optimism →Behavioral Intention | PU | 0.174 | 0.071 | 0.303 | 0.003 |
| Innovativeness → Behavioral Intention | PEU | 0.023 | 0.001 | 0.063 | 0.036 |
| Innovativeness → Perceived Usefulness | PEU | 0.081 | 0.012 | 0.168 | 0.021 |
| Perceived Ease of Use →BI | PU | 0.417 | 0.271 | 0.634 | < 0.001 |

Source: AMOS analysis output

PEU: perceived ease of use; PU: Perceived usefulness; BI: Behavioral intention; CI: Confidence Interval

*"There is an assigned team to offer swift internet access service and technical support within six hours from the time of the customer's request. However, customers are unable to afford the subscription, service, and related fees." (Regional Level Supervisor, 8 years of working experience)*

Similarly, the majority of the informants agreed that most of the professionals had better awareness of the need for technology. Informants added that the era is an ideal opportunity to adopt teleradiology; however, a lack of adequate budget and infrastructure is still a major challenge for hospitals.

*"We are currently using a 6MB-speed internet connection, which might not be enough to support high-bandwidth-demanding eHealth technologies like teleradiology. We have a plan to upgrade our internet bandwidth to a better bandwidth, but a shortage of budget is our current critical problem." (Hospital Manager, 5 years of working experience)*

Another informant added:

*"We are very eager to see the implementation of teleradiology. This is the area that needs more attention since patients are still suffering a lot to get an appropriate radiology service. We have tried different options to solve the problem, like bringing radiologists from other referral hospitals for a short period of time to give an onsite radiology service, but budget is our major constraint." (Hospital Manager, 4 years of working experience)*

Aside from funding concerns, frequent power outages and a lack of technical experience have been identified as bottlenecks for the health sector to introduce new eHealth technologies like teleradiology. An informant from the regional health bureau stated:

*"Surprisingly, power might not be available for the whole day, or even days or weeks. As a result, hospitals use generators as a backup power source, but it requires a large budget for fuel."*

The key informant also added:

*"In some hospitals, there are IT professionals at diploma level who are not technically capable of maintaining digital imaging equipment. Even sometimes, the technical problems of the*

*digital imaging machines might not be solved by the regional health bureau's biomedical engineers." (Technical Expert Team Leader, 10 years of working experience)*

Moreover, assessing the level of technology acceptance, enhancing end-users' commitment, supportive supervision, and close follow-up are very important for technology adoption. A key informant from the federal ministry of health stated:

*"Scientifically we know that the level of healthcare professionals' technology acceptance will not be the same. There are early adopters, innovators, and laggards. Most of them accept the technology, but lack of training, lack of familiarity with the technology and lack of commitment are some of the reasons to lately accept technologies. Even people who have got training might not be committed to using technologies unless there is close follow-up." (System Administrator and Technical Advisor, 15 years of working experience)*

## Discussion

This study examined the effect of technology readiness on health professional teleradiology acceptance in the Amhara region state public hospitals. Results indicated that more than three-fourths of health professionals have had the intention to use teleradiology. Optimism and innovativeness from the technology readiness dimensions, as well as perceived ease of use and perceived usefulness from the technology acceptance model dimension, influenced health professionals' intentions to use teleradiology.

This study shows a higher proportion of health professionals' readiness towards teleradiology acceptance compared to previous studies conducted in Ethiopia [14–16, 61, 62]. The difference could be attributed to the outcome measurement. The current study computed the composite mean score and dichotomized the composite score using the cutoff point 3 (by merging the lowest three into one category, 'not ready', and the remaining two into the second category 'ready'). However, others used mean, median, and other cut points to categorize the composite score which could ultimately affect the proportion of readiness [16, 61]. The higher proportion of readiness among health professionals towards teleradiology acceptance in Ethiopia compared to previous studies suggests positive progress and increased openness to adopting this technology. It indicates that health professionals are prepared and willing to embrace teleradiology, leading to improved efficiency, accessibility, and quality of radiology services in the country.

Moreover, the analysis of this study also suggests that more than three-quarters of health professionals have shown an intention to use teleradiology. We observed that the proportion of intention to use is higher than the proportion of readiness. This implies that when intention exceeds readiness, it emphasizes the need to address knowledge gaps, improve infrastructure, manage change effectively, and overcome implementation challenges. This bridges the gap and enables successful adoption and utilization of the technology. However, this finding is lower than a study conducted in Taiwan [63]. The possible explanation could be health professionals' previous exposure to new healthcare technologies. The current study involved participants who had limited practical experience in teleradiology during their graduate and undergraduate classes. In contrast, the study conducted in Taiwan included nurse professionals who likely had more prior exposure to healthcare technology use. Factors such as government support, training opportunities, technological advancements, and accessibility could also influence the intention to use. Moreover, practical experience influences health professionals' adoption of teleradiology. A study done in the United States found that 48.4% of teleradiology challenges were related to proximity to technology [64].

The test results supported six out of the eleven proposed hypotheses. These supported hypotheses include: Optimism positively influenced the perceived ease of use and perceived usefulness of teleradiology (H1 and H2). Innovativeness positively influenced the perceived ease of use of teleradiology (H3). Additionally, perceived ease of use positively influenced the perceived usefulness and behavior intention in using teleradiology (H9 and H10). Furthermore, perceived usefulness positively influenced the behavior intention in using teleradiology (H11).

## Effect of optimism on perceived ease of use and perceived usefulness

Optimism contributes positively toward perceived ease of use and perceived usefulness. The study findings are in agreement with a prior study that was conducted in Taiwan [18]. The positive relationship between optimism and the cognitive dimensions of TAM implies that the more optimistic the professionals are, the more they view technology as simple to use, more useful, and enabling them to develop self-confidence in their future work [40, 65]. Similarly, optimism shows a significant positive indirect effect on perceived usefulness, mediated by perceived ease of use. Techno-optimist professionals consider the technology useful only when the system is easy to use [66]. This finding suggests that optimistic individuals find teleradiology easier to use, leading to a stronger belief in its usefulness. Furthermore, it also highlights that optimism plays a role in shaping attitudes and beliefs about teleradiology's usability and benefits. Hence, promoting optimism among users could enhance acceptance and adoption of teleradiology systems.

Moreover, a significant positive indirect effect was observed between optimism and behavioral intention, mediated by perceived usefulness. The possible explanation for this could be that optimist professionals assume that technology plays a key role in ensuring that the good prevails over the bad [66]. This highlights the importance of optimism in shaping attitudes toward teleradiology. Thus, strategies promoting optimism could enhance users' intentions to adopt and utilize teleradiology systems.

## Effect of innovativeness on perceived ease of use and perceived usefulness

The present study result demonstrates a significant positive influence of innovativeness on the perceived ease of use of teleradiology, which is similar to a study conducted in Taiwan [18]. This could be explained by the fact that highly innovative people are assumed to be more ready to accept and test emerging technologies than less innovative people [29]. This implies that individuals who are more innovative find teleradiology technology easier to use. This highlights how embracing new and innovative solutions positively influences their perception of teleradiology's ease of use. However, in our findings, innovativeness does not show a direct and significant influence on health professionals' perceived usefulness of teleradiology unless mediated by perceived ease of use. This highlights that innovative people are more cautious of technology because they are aware of the latest innovations and believe that technology is useful only when it is simple to use [40].

Furthermore, innovativeness showed a positive statistically significant indirect effect on behavioral intention when mediated by perceived ease of use. This implies that innovative people intend to use new technologies more when they feel that technology is easy to use. Otherwise, the paucity of understanding of technology inhibits end-users' exposure to new technologies. Thus, fostering an innovative mindset among users can enhance the adoption of teleradiology systems.

### Effect of perceived ease of use on perceived usefulness

The data for this study indicates that perceived ease of use has a direct positive influence on behavioral intention to use teleradiology. Similarly, perceived ease of use shows a statistically significant effect on the perceived usefulness of teleradiology. This finding is similar to previous studies conducted in Ethiopia [67] and elsewhere abroad in Taiwan [36] and India [68]. The possible explanation for this could be that the more user-friendly a technology is, the higher its perceived usefulness value among individuals [27, 32]. This indicates that perceived ease of use determines perceived usefulness, and a difficult-to-use technology may fail to attract end-users' attention.

### Effect of perceived ease of use and perceived usefulness on behavioral intention

The results demonstrate that cognitive factors (perceived ease of use and perceived usefulness) positively influence health professionals' behavioral intentions to use teleradiology. Our study demonstrates that perceived usefulness and ease of use encourage behavioral intention to use teleradiology, which is similar to previous research studies [18, 68]. This implies that when professionals perceive a technology is simple to use, helpful to their future work, and whose benefits outweigh the drawbacks [69], they might have a positive intention towards the use of the system in the future. Moreover, our result shows a positive relationship between perceived ease of use and the behavioral intention of health professionals to use teleradiology through perceived usefulness. Furthermore, the indirect effect is almost three times higher than the direct effect ($\beta = 0.42$ vs. $\beta = 0.16$ respectively). When the indirect effect is higher than the direct effect, it suggests that the mediating variable has a stronger influence on the outcome variable. This highlights the importance of considering the mediating process and understanding the underlying mechanisms through which the independent variable affects the outcome variable. Our study finding is in consistent with a previous study done in Poland [70].

However, the readiness measuring constructs of discomfort and insecurity do not show any direct or indirect statistical significance on perceived usefulness, perceived ease of use, and behavioral intention to use teleradiology. This could suggest that these factors are not significant barriers to the adoption of teleradiology.

### Limitations of the study

Our findings should be interpreted in the context of the study's limitations, regardless of their theoretical and practical implications. Despite the fact that many publications recommend a 10:1 free parameter estimate to determine the minimum sample size for structural equation modeling, we used the thumb rule of 5:1 due to the limited number of health professionals available in the selected study area. However, to maximize the final sample size, we considered all the eligible health professionals.

Moreover, since the study used self-reported data, the possibility of recall bias was considerable. However, as a strategy to reduce bias, we used a pilot-tested data collection tool. Furthermore, this study used TRAM to evaluate the effect of technological readiness on health professionals' technology acceptance; however, this model lacks the ability to generate evidence related to technological context factors (performance expectancy and effort expectancy) [31]. As a result, it is difficult to generalize the level of acceptance of health professionals.

### Conclusions

We found a significantly higher proportion of health professionals' technology readiness and intention to use teleradiology. In total, the model explained 54% of the variation in behavioral

intention to use teleradiology, indicating that the model was able to explain a substantial portion of the decision-making process. This study found that perceived ease of use and perceived usefulness showed a direct effect on health professionals' intention to use teleradiology. Notably, perceived ease of use and perceived usefulness directly influenced health professionals' intentions to use teleradiology. Additionally, factors such as optimism, innovativeness, and perceived ease of use indirectly influenced their intentions to adopt teleradiology. Furthermore, we also found that low bandwidth of internet connection, frequent power interruption, shortage of budget, and lack of technical experts were major contributors to implementing teleradiology. These findings highlight the importance of addressing these challenges before the actual implementation of teleradiology. Specifically, ensuring the user-friendliness of the system, improving infrastructure, adequate budget allocation, and capacity-building activities are very crucial steps to facilitate the successful adoption and utilization of teleradiology. Based on our results, we recommend further research that takes into account external factors to enhance the predictive capability of the proposed model. Considering additional variables and contextual factors will provide a more comprehensive understanding of the factors influencing health professionals' intention to adopt and utilize teleradiology, ultimately contributing to the improvement of its implementation and effectiveness in healthcare settings.

## Supporting information

**S1 Data set. Data set of health professionals' technology readiness on the acceptance of teleradiology in the Amhara regional state public hospitals, northwest Ethiopia: Using technology readiness acceptance model.**
(XLSX)

**S1 Audio. Audio record for participant one.**
(MP3)

**S2 Audio. Audio record for participant two.**
(MP3)

**S3 Audio. Audio record for participant three.**
(MP3)

**S4 Audio. Audio record for participant four.**
(MP3)

**S5 Audio. Audio record for participant five.**
(M4A)

**S6 Audio. Audio record for participant six.**
(MP4)

**S7 Audio. Audio record for participant seven.**
(MP3)

**S8 Audio. Audio record for participant eight.**
(MP3)

## Acknowledgments

We would like to thank the University of Gondar for providing ethical clearance to conduct this study. The authors are grateful to hospitals, data collectors, supervisors, study participants, and other stakeholders involved in this study. We would also like to thank the University of

South-Eastern Norway for offering the student exchange opportunity and the NURTURE project for covering the transportation and accommodation costs.

## Author Contributions

**Conceptualization:** Araya Mesfin Nigatu, Tesfahun Melese Yilma, Lemma Derseh Gezie, Yonathan Gebrewold, Binyam Tilahun.

**Data curation:** Araya Mesfin Nigatu, Tesfahun Melese Yilma, Binyam Tilahun.

**Formal analysis:** Araya Mesfin Nigatu, Binyam Tilahun.

**Funding acquisition:** Araya Mesfin Nigatu.

**Investigation:** Araya Mesfin Nigatu, Lemma Derseh Gezie, Yonathan Gebrewold, Monika Knudsen Gullslett, Shegaw Anagaw Mengiste, Binyam Tilahun.

**Methodology:** Araya Mesfin Nigatu, Lemma Derseh Gezie, Yonathan Gebrewold, Monika Knudsen Gullslett, Shegaw Anagaw Mengiste, Binyam Tilahun.

**Project administration:** Araya Mesfin Nigatu.

**Resources:** Araya Mesfin Nigatu, Binyam Tilahun.

**Software:** Araya Mesfin Nigatu.

**Supervision:** Araya Mesfin Nigatu, Yonathan Gebrewold, Monika Knudsen Gullslett, Shegaw Anagaw Mengiste, Binyam Tilahun.

**Validation:** Araya Mesfin Nigatu.

**Visualization:** Araya Mesfin Nigatu, Shegaw Anagaw Mengiste.

**Writing – original draft:** Araya Mesfin Nigatu, Tesfahun Melese Yilma.

**Writing – review & editing:** Araya Mesfin Nigatu, Tesfahun Melese Yilma, Lemma Derseh Gezie, Yonathan Gebrewold, Monika Knudsen Gullslett, Shegaw Anagaw Mengiste, Binyam Tilahun.

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
