## [Decision Letter · Decision Letter 0]

20 Sep 2023

PONE-D-23-10534Health professionals’ technology readiness on the acceptance of teleradiology in the Amhara regional state public hospitals, northwest Ethiopia: using technology readiness acceptance model (TRAM)PLOS ONE

Dear Dr. Nigatu

Thank you for submitting your manuscript to PLOS ONE. After careful consideration, we feel that it has merit but does not fully meet PLOS ONE’s publication criteria as it currently stands. Therefore, we invite you to submit a revised version of the manuscript that addresses the points raised during the review process.

We look forward to receiving your revised manuscript.

Kind regards,

Najmul Hasan, PhD

Academic Editor

PLOS ONE

Journal Requirements:

“The work was supported financially by the University of Gondar under award number (Ref. No: 05/2886/2013). The funder, however, played no role in the study design, data collections, analysis of results, interpretation of data, writing of the manuscript, nor the decision to submit the manuscript for publication.”

“We would like to thank the University of Gondar for providing the fund and Ethical clearance to conduct this study.”

“The work was supported financially by the University of Gondar under award number (Ref. No: 05/2886/2013). The funder, however, played no role in the study design, data collections, analysis of results, interpretation of data, writing of the manuscript, nor the decision to submit the manuscript for publication.”

Additional Editor Comments:

The paper has merit. However, before recommending the publication, I suggest addressing the following issues.

The introduction lacks lackings of proper identification of research questions. Even the authors did not provide clear logic as to why this study is important.

My suggestion is to follow the following steps

Background of the study > importance / main motivation of the study > properly formulate the research question and be specific on research objective > prospective contribution and finally, the introduction will finish with the "structure of the manuscript".

While choosing the theoretical model looks good, the logic behind constructing the hypothesis lacks.

The methodology section also looks good. However, I am not satisfied - why qualitative component needed after quantitative evidence? What is the main motivation for studying the qualitative component?

Does the discussion section provide sufficient information on the research questions? (though research questions have lacked).

There is a lack of both theoretical and practical contributions. For example, How does this study contribute to the scholarly world or extend the scope of existing literature? How does this study contribute to the health professionals in a specific manner? How would this study be beneficial for Ethiopian public and private healthcare systems/industry?

Reviewers' comments:

Reviewer's Responses to Questions

**Comments to the Author**

1. Is the manuscript technically sound, and do the data support the conclusions?

Reviewer #1: Yes

2. Has the statistical analysis been performed appropriately and rigorously? 

Reviewer #1: Yes

3. Have the authors made all data underlying the findings in their manuscript fully available?

Reviewer #1: Yes

4. Is the manuscript presented in an intelligible fashion and written in standard English?

Reviewer #1: Yes

5. Review Comments to the Author

Reviewer #1: The paper is well articulated and it is timely. Conducting such types of studies “Health professionals’ technology readiness on the acceptance of teleradiology” will enhance the telemedicine penetration and improve the health service coverage by increasing health access of the individuals. Over all the manuscript is well written, but I have some questions and comments that have to be addressed by the authors to improve the quality of the paper.

1. It would be better to add some information on the Telemedicine/Teleradiology practice in Amhara region of Ethiopia, even pilot implementations.

2. Some of the questions in the model assessment you had asked Perceived Ease of Use. Do you think it is easy to answer this question without having a single exposure on Telemedicine before?

3. For the qualitative part you only asked eight key informants from different sectors, do you think it is sufficient? Did the data saturate at the 8th key informant? Make it clear.

4. In reporting the qualitative finding rather than saying he/she it is better to describe the respondent keeping them anonymous.

6. PLOS authors have the option to publish the peer review history of their article (what does this mean?). If published, this will include your full peer review and any attached files.

Reviewer #1: No

---

## [Author Response · Author response to Decision Letter 0]

23 Jan 2024

Thank you reviwer for your relevant questions. We have tried to address all the questions and uploaded a point by point response.

---

## [Decision Letter · Decision Letter 1]

11 Mar 2024

Health professionals’ technology readiness on the acceptance of teleradiology in the Amhara regional state public hospitals, northwest Ethiopia: using technology readiness acceptance model (TRAM)

PONE-D-23-10534R1

Dear Dr. Nigatu

We’re pleased to inform you that your manuscript has been judged scientifically suitable for publication and will be formally accepted for publication once it meets all outstanding technical requirements.

Kind regards,

Najmul Hasan, PhD

Academic Editor

PLOS ONE

Additional Editor Comments (optional):

Reviewers' comments:

Reviewer's Responses to Questions

**Comments to the Author**

1. If the authors have adequately addressed your comments raised in a previous round of review and you feel that this manuscript is now acceptable for publication, you may indicate that here to bypass the “Comments to the Author” section, enter your conflict of interest statement in the “Confidential to Editor” section, and submit your "Accept" recommendation.

Reviewer #1: All comments have been addressed

Reviewer #2: All comments have been addressed

2. Is the manuscript technically sound, and do the data support the conclusions?

Reviewer #1: Yes

Reviewer #2: Yes

3. Has the statistical analysis been performed appropriately and rigorously? 

Reviewer #1: Yes

Reviewer #2: Yes

4. Have the authors made all data underlying the findings in their manuscript fully available?

Reviewer #1: Yes

Reviewer #2: Yes

5. Is the manuscript presented in an intelligible fashion and written in standard English?

Reviewer #1: Yes

Reviewer #2: Yes

6. Review Comments to the Author

Reviewer #1: All the comments are addressed by the author. The conclusion has to be a little bit shorter and precise, repeating what is presented in the result section is not that much important.

Reviewer #2: Strength

• The research paper is well written, well organized and the authors raised currently the most relevant and novel topic in Ethiopia.

Comments

1. Page 12, the 1st sentence in the data collection procedures and quality control measures section said: The qualitative data were collected using a self-administered and structured pre-tested Amharic version questionnaire. Could you want to mean the quantitative?

2. Page 16, the 1st sentence in the Health Professions’ eHealth Literacy section said: The majority of the health professionals 334 61.1%)

So try to correct use of brackets like 334 (61.1%)

7. PLOS authors have the option to publish the peer review history of their article (what does this mean?). If published, this will include your full peer review and any attached files.

Reviewer #1: No

Reviewer #2: No

---

## [Editor Report · Acceptance letter]

19 Mar 2024

PONE-D-23-10534R1 

PLOS ONE

Dear Dr. Nigatu, 

I'm pleased to inform you that your manuscript has been deemed suitable for publication in PLOS ONE. Congratulations! Your manuscript is now being handed over to our production team.

Kind regards, 

on behalf of

Dr. Najmul Hasan 

Academic Editor

PLOS ONE